# Towards Better Generalization in Lifelong Person Re-Identification with Flatness-Aware Learning

## Abstract

Lifelong person re-identification (LReID) requires models to continuously learn from sequentially arriving domains while retaining discriminative power for previously seen identities. A key challenge is to prevent catastrophic forgetting without access to old data, especially under exemplar-free constraints. In this paper, we propose a novel LReID method that unifies selective flatness-aware optimization, dual-model training, and model interpolation. Specifically, we maintain two separate models per task: a stability model trained with the distillation loss to retain the prior knowledge, and a plasticity model optimized solely for the current domain. To improve the performance of generalization and retention, we selectively apply Sharpness-Aware Minimization (SAM) only to the distillation loss, guiding the stability model toward flat and robust solutions. After task-specific training, these two models are fused through weight-space interpolation, producing a single model that balances stability and adaptability. The resulting model is used to initialize both branches for the next task, enabling continual knowledge integration. Our method is lightweight, modular, and readily compatible with existing LReID frameworks. Extensive experimental results consistently demonstrate that the proposed flat-minima-guided model fusion strategy consistently improves the overall performance of LReID.

## 1 Introduction

Person re-identification (ReID) is a fundamental task in computer vision that aims to match individuals across camera views under varying conditions. With its growing importance in surveillance, security, and retrieval applications, deep learning-based ReID has achieved substantial progress. Recently, lifelong person re-identification (LReID) has emerged as a more practical paradigm, in which models must sequentially adapt to new domains without access to previously seen data. ReID is inherently an open-set problem, as the test identities differ from those observed during training. This necessitates a strong generalization capability across continually arriving domains. To further address privacy concerns, recent methods operate under exemplar-free constraints (Pu et al., 2021; Sun & Mu, 2022; Xu et al., 2024a; 2025a), where data from past domains cannot be stored. The main challenge lies in acquiring new knowledge without catastrophically forgetting prior representations. This creates a tension between preserving previous knowledge and adapting to new tasks, a problem that current methods still struggle to resolve.

Most existing LReID methods (Wu & Gong, 2021; Pu et al., 2021; Sun & Mu, 2022; Xu et al., 2024b) address this challenge using regularization-based strategies, primarily via distillation losses or weight-space constraints. A widely adopted approach maintains both old and new models during training and applies a distillation loss to align the new model's outputs with those of the old model. This paradigm has become standard in LReID, with recent extensions proposing refined variants such as relation-aware (Xu et al., 2024b) and bi-directional (Yu et al., 2023) distillation to enhance knowledge transfer. However, because these methods only constrain the new model to align with the old one, they do not guide optimization toward more stable or robust regions of the loss landscape. As a result, the model can converge to sharp or overly specialized solutions, where small parameter updates lead to large changes in predictions or poor transferability across domains. Such sensitivity hinders effective consolidation of knowledge over multiple tasks.

As illustrated in Figure 1, these approaches are aligned with regularization-based continual learning methods, such as knowledge distillation (Figure 1a), where the updates are constrained to remain close to the previous solution $\theta$. While this helps preserve prior knowledge, it often limits effective adaptation to new tasks. Moreover, such methods lack explicit control over how parameter updates contribute to forgetting and do not actively encourage generalization, which is crucial in ReID where test identities differ from those seen

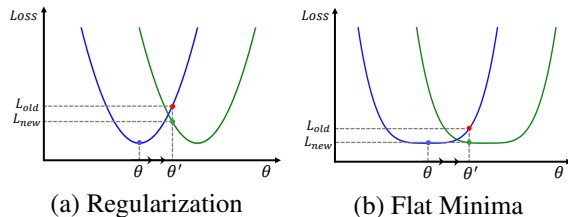

(a) Regularization          (b) Flat Minima

Figure 1: Illustration of two continual learning approaches, focusing on parameter updates' effect on the loss landscape. The blue curve represents the previous task, and the green curve the new task.

during training. In contrast, recent continual learning (CL) research advocates an optimization-guided perspective that steers solutions toward flatter minima (Figure 1b). Flat minima-based methods search for flatter regions in the loss landscape to achieve a better balance between stability and plasticity across tasks, thereby improving both knowledge preservation and generalization.

Following this idea, many studies (Shi et al., 2021; Deng et al., 2021; Bian et al., 2024) have demonstrated the benefits of flattening the loss landscape in CL. Solutions converging to flat minima tend to generalize better and are more robust to parameter perturbations, making them well-suited for continual learning. Sharpness-Aware Minimization (SAM) (Foret et al., 2020) has been widely used for this purpose, showing promising results across various CL settings (Yang et al., 2023; Tran Tung et al., 2023). However, existing methods often apply SAM uniformly to all loss components or rely on global strategies, such as gradient alignment, to mitigate task interference, without considering the different objectives of each loss. As shown in Figure 2, SAM produces flatter loss landscapes than SGD, particularly after multiple tasks, and this property correlates with improved generalization and reduced forgetting. Similarly, Figure 3 shows that SAM maintains stable performance on previous domains while improving generalization to unseen domains compared to standard SGD. Despite these advances, no prior work in LReID or general continual learning has explored selectively applying SAM, especially to the distillation loss, to robustly preserve previous knowledge.

In this paper, we propose a novel method for LReID that unifies three core components: (1) selective flatness-aware optimization, (2) dual-model training with decoupled objectives, and (3) weight-space interpolation for model fusion. Instead of training a single model, we maintain two: a *plasticity model* optimized on the current domain, and a *stability model* trained with distillation loss to preserve prior knowledge. We selectively apply SAM only to the distillation loss of the stability model, ensuring its convergence to flat and resilient regions while leaving the plasticity model free to explore task-specific solutions. After training, the two models are linearly interpolated in parameter space, producing a single fused model that captures both adaptability and retention. This fused model then serves as the initialization for the next task, enabling smooth knowledge transition.

To the best of our knowledge, this is the first approach in LReID and general continual learning to combine selective SAM, dual-model training, and flat-minima-guided interpolation within a unified framework. Our method is lightweight, modular, and easily integrable into existing distillation-based pipelines. Extensive experiments show consistent improvements on seen domains and enhanced generalization to unseen domains, going beyond the conventional stability–plasticity trade-off.

## 2 RELATED WORK

### 2.1 PERSON RE-IDENTIFICATION

ReID has been widely studied, achieving high performance on large-scale benchmarks (Zheng et al., 2015; Wei et al., 2018). To address limitations in pose diversity, Liu et al. (2018) proposed a data augmentation framework that enhances the model's discriminative ability via synthesized pose variations. In addition, Li et al. (2018) introduced a harmonious attention network that jointly learns pixel-wise and part-wise attention to obtain robust features under misalignment and pose variation. UDA-based approaches (Mekhazni et al., 2020) have also been explored, aiming to transfer knowledge from a labeled source domain to an unlabeled target domain by aligning pairwise distance distributions in the dissimilarity space. Furthermore, fully unsupervised methods (Chen et al., 2021;

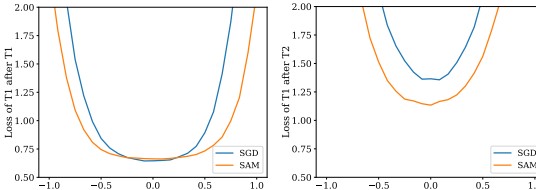 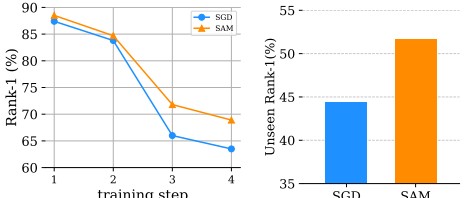

Figure 2: Loss landscape of task 1 (Market1501) after learning (left) task 1 and (right) task 2 (CUHK-SYSU), using SGD and SAM optimizers under a ReID setting (details in Appendix I).

Figure 3: Evaluation of the non-forgetting performance on the first dataset at each training step (left), and performance on unseen datasets (right).

Dai et al., 2022) often rely on clustering-based pseudo-labeling and contrastive learning to train robust models without any annotations. However, all these methods operate under the assumption that the entire training data is available at once, which is often unrealistic in real-world scenarios where data arrives continuously.

## 2.2 LIFELONG PERSON RE-IDENTIFICATION

LReID aims to incrementally adapt to new domains without forgetting previously learned knowledge. Most prior works utilize knowledge distillation to alleviate catastrophic forgetting (Pu et al., 2021; Sun & Mu, 2022; Xu et al., 2024b). Pu et al. (2021) introduced the first LReID method, AKA, which encodes dynamic class relationships across domains via an accumulated knowledge graph. Sun & Mu (2022) proposed PatchKD, a patch-wise distillation approach that enforces local feature consistency instead of global alignment, mitigating spatial misalignment and enhancing generalization. Xu et al. (2024b) further refined this idea through LSTKC, which introduces a relation matrix based on instance similarity and label supervision to reduce relation errors and domain gaps.

Replay-based approaches (Ge et al., 2022; Yu et al., 2023) store a small number of old samples to preserve previous knowledge. Ge et al. (2022) proposed PTKP, which enhances replay efficiency by projecting new data into the feature space of past domains. Yu et al. (2023) introduced KRKC, a bidirectional distillation framework that encourages mutual knowledge transfer between the old and new models. More recently, exemplar-free methods have emerged. Xu et al. (2024a) proposed DKP, which stores distribution-aware class prototypes from previous domains to enable exemplar-free distillation and enhance model stability. Xu et al. (2025b) introduced LSTKC++, which decomposes the old model into long-term and short-term components and applies temporally-aware distillation that assigns greater importance to earlier tasks based on their temporal order.

In contrast to prior work, our method introduces a plug-in framework that does not require exemplar storage and can be readily integrated into existing LReID frameworks. Unlike existing approaches that rely on a single model, we decouple stability and plasticity through two models optimized with distillation and ReID losses, respectively. After training, the models are interpolated in parameter space to integrate knowledge. To further enhance robustness, we selectively apply sharpness-aware optimization to enhance convergence to flat minima, which are known to offer better generalization and resilience in continual learning settings.

## 2.3 FLAT MINIMA IN CONTINUAL LEARNING

Flat minima-based approaches are increasingly explored in CL due to their robustness to perturbations and ability to preserve knowledge across tasks. Shi et al. (2021) demonstrated in incremental few-shot learning that locating a flat minimum during initial training facilitates the retention of prior knowledge by allowing subsequent tasks to be fine-tuned within the same region.

Subsequent works have explicitly incorporated flatness into CL frameworks. Tran Tung et al. (2023) integrated SAM (Foret et al., 2020) into replay-based CL, encouraging convergence to flatter solutions and addressing multi-objective gradient conflicts through aggregated SAM gradients across tasks. Li et al. (2024b) proposed UniGrad-FS, which unifies gradient projection and flatness optimization by perturbing model weights to explore wide minima and projecting gradients into low-conflict subspaces, leading to more stable task transitions. Huang et al. (2025) extended this

principle to continual semantic segmentation with AlterSGD, which alternates descent and ascent steps to efficiently converge toward flat minima and reduce forgetting. Most recently, Bian et al. (2024) formally decomposed forgetting into interference and forgetting-induced sharpness, introducing flatness-preserving consistency regularization and Hessian-guided smoothing to avoid sharp shifts in the solution space. This framework not only mitigates forgetting but also enforces smooth transitions between tasks.

To this end, our method incorporates flatness guidance into the LReID process. SAM is selectively applied to the distillation loss of the stability model, encouraging convergence to robust flat minima without compromising adaptability. Furthermore, weight-space interpolation is performed after each task to preserve shared flat regions, facilitating smooth knowledge integration. These strategies align with the flat-minima principle, jointly enhancing retention and generalization in a CL setting.

## 3 PRELIMINARY

### 3.1 LIFELONG PERSON RE-IDENTIFICATION

LReID aims to train a model that sequentially learns from multiple datasets without accessing previous data, while preserving discriminative power across all learned domains. Formally, let $\mathcal{D}_1, \mathcal{D}_2, \ldots, \mathcal{D}_T$ be a sequence of datasets, each with distinct person identities. At each stage $t$, a model $\theta_t$ is optimized only on $\mathcal{D}_t$, but its performance is evaluated across all datasets $\{\mathcal{D}_1, \ldots, \mathcal{D}_t\}$.

To tackle catastrophic forgetting, knowledge distillation (KD) has been widely adopted. During training on $\mathcal{D}_t$, the model $\theta_t$ is encouraged to preserve the knowledge of the previously trained model $\theta_{t-1}$ using a distillation loss. Various formulations of distillation loss are possible, including cross-entropy, KL divergence, and L2 distance at the logits level, as well as L2- and cosine similarity-based alignment losses at the feature level (Li et al., 2024a). Among these, we adopt the cosine similarity-based loss, which effectively preserves the directional alignment of instance features:

$$\mathcal{L}_{\text{KD}} = \frac{1}{B} \sum_{i=1}^{B} \left(1 - \cos\left(f_t(x_i), f_{t-1}(x_i)\right)\right),\tag{1}$$

where $f_t(x_i)$ and $f_{t-1}(x_i)$ denote the features of sample $x_i$ extracted by $\theta_t$ and $\theta_{t-1}$, respectively, and $B$ is the batch size.

Alternatively, relation matrix-based approaches preserve knowledge by aligning pairwise similarity patterns within a batch (Xu et al., 2024b), thereby effectively capturing the structural relationships between features through their cosine similarities. These approaches offer complementary strengths depending on whether the focus lies in instance-level correspondence, structural consistency, or output-level alignment.

The model is also trained using the ReID loss, which consists of cross-entropy loss and triplet loss, denoted by $\mathcal{L}_{\text{ReID}}$. These losses are essential for learning identity discriminative features on the current domain and are widely adopted as baseline objectives in previous LReID studies (Sun & Mu, 2022; Xu et al., 2024b). The overall training objective becomes to minimize the total loss:

$$\mathcal{L}_{\text{total}} = \mathcal{L}_{\text{ReID}} + \mathcal{L}_{\text{KD}}.\tag{2}$$

### 3.2 SHARPNESS-AWARE MINIMIZATION

SAM (Foret et al., 2020) is an optimization technique designed to guide models toward flat minima by minimizing the worst-case loss within a neighborhood of the current parameters. The motivation behind SAM is to avoid sharp local minima that are highly sensitive to small perturbations in the weights, thereby improving generalization.

In SAM, a perturbation vector $\epsilon$ is computed in the direction that maximizes the training loss within an $\ell_2$-ball centered at the current parameters (see Appendix B for the full formulation):

$$\epsilon = \rho \cdot \frac{\nabla_w \mathcal{L}(w)}{\|\nabla_w \mathcal{L}(w)\|_2},\tag{3}$$

where $\rho$ is the perturbation radius, and $\nabla_w \mathcal{L}(w)$ is the gradient of the loss with respect to the model weights $w$.

Once the perturbation is obtained, SAM updates the weights using the gradient computed at the perturbed location:

$$w \leftarrow w - \alpha \cdot \nabla_w \mathcal{L}(w + \epsilon), \tag{4}$$

where $\alpha$ denotes the learning rate. This two-step update encourages the model to converge to flatter and more generalizable minima. For a more detailed explanation of Adaptive Sharpness-Aware Minimization (ASAM), including the relevant formulation and perturbation computation, please refer to Appendix B.

In our method, the SAM update rule itself remains unchanged. The only modification lies in how the perturbation is computed for each branch. The stability branch derives its perturbation from the ReID loss, while the plasticity branch computes it from the distillation loss. This separation forms the basis of selective SAM and enables each branch to optimize along its intended direction within the unified framework.

## 4 PROPOSED METHOD

We propose a plug-and-play framework for LReID that separates the optimization for plasticity and stability by training two models with selective perturbations. As illustrated in Figure 5, our method follows a two-phase training procedure for each task, and the detailed steps are provided in Algorithm 1. The first phase involves initializing the models with ReID-oriented SAM training. In our method, we adopt ASAM rather than standard SAM, as the adaptive normalization term leads to more balanced perturbations across layers with heterogeneous scales. This proves particularly beneficial in our dual-model setting, where different loss components guide the optimization of the sta-

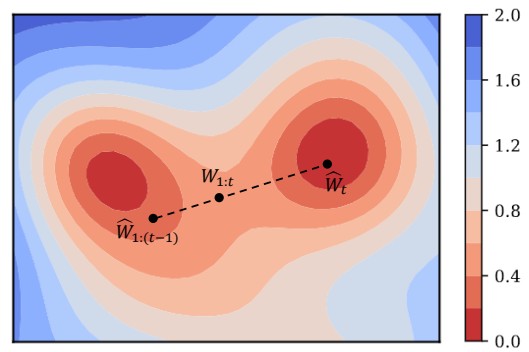

Figure 4: Loss surface of the current task $t$, illustrating the interpolation between the stability model and the plasticity model to obtain the fused model.

bility and plasticity branches independently. The following formulation is based on standard SAM for simplicity, but all equations remain valid under ASAM by replacing the perturbation accordingly. We empirically validate this choice in Appendix C, where ASAM shows improved retention and generalization over SAM. After training, the stability and plasticity models are fused via weight-space interpolation, combining adaptability and retention for smooth knowledge transfer.

A key motivation behind this design is that a single model cannot satisfy the conflicting objectives of LReID. Gradients from the current domain and those from past domains often interfere, causing either forgetting or limited adaptation. To address this, we integrate three complementary components into a unified framework. Selective SAM guides the stability branch toward flatter solutions by applying perturbations only to the distillation loss, dual-model training separates the roles of plasticity and stability, and weight-space interpolation merges their trajectories into a balanced model for the next task. Together, these components alleviate gradient interference and enable stable knowledge transfer across domains.

### 4.1 TASK 1: INITIALIZATION WITH REID-ORIENTED SAM TRAINING

For the first task $T_1$, we initialize the model parameters $W_{1:1}$ randomly and train the model using SAM with the ReID loss $\mathcal{L}_{\text{ReID}}$ only. Specifically, at each epoch, the SAM perturbation $\epsilon$ is computed using the gradient of ReID loss:

$$\epsilon = \rho \cdot \frac{\nabla \mathcal{L}_{\text{ReID}}(W)}{\|\nabla \mathcal{L}_{\text{ReID}}(W)\|_2}, \tag{5}$$

and the model is updated with the perturbed gradient:

$$W \leftarrow W - \alpha \cdot \nabla \mathcal{L}_{\text{ReID}}(W + \epsilon). \tag{6}$$

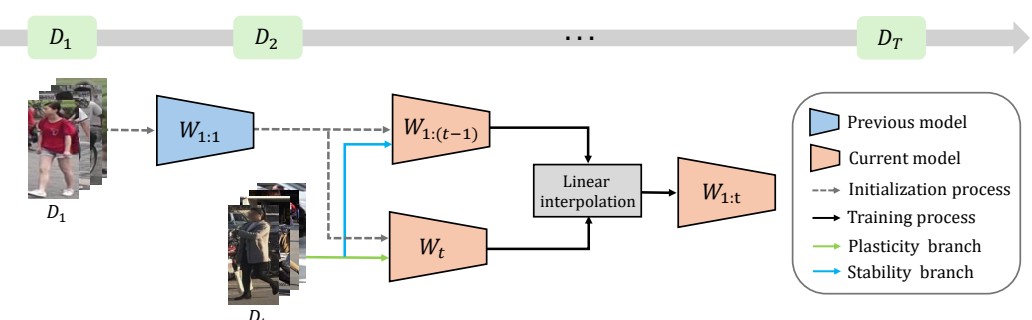

Figure 5: Overview of our framework.

By applying SAM in the first task, the model is guided to converge to flat and more generalizable solutions for the ReID objective. This leads to improved generalization performance and establishes a more reliable starting point for continual learning.

### 4.2 TASK $t \geq 2$: DUAL-MODEL SELECTIVE SAM TRAINING

From the second task, we employ two separate models:

- $\widehat{W}_{1:(t-1)}$: a stability branch updated via SAM using the distillation loss $\mathcal{L}_{\text{KD}}$
- $\widehat{W}_t$: a plasticity branch updated via SAM using the ReID loss $\mathcal{L}_{\text{ReID}}$

At each epoch, both models are perturbed using SAM. The perturbation for the stability branch is:

$$\epsilon_{\text{KD}} = \rho \cdot \frac{\nabla \mathcal{L}_{\text{KD}}}{\|\nabla \mathcal{L}_{\text{KD}}\|_2}, \tag{7}$$

and the perturbation for the plasticity branch is similarly:

$$\epsilon_{\text{ReID}} = \rho \cdot \frac{\nabla \mathcal{L}_{\text{ReID}}}{\|\nabla \mathcal{L}_{\text{ReID}}\|_2}. \tag{8}$$

Note that the perturbations $\epsilon_{\text{KD}}$ and $\epsilon_{\text{ReID}}$ are computed from different objectives. This selective perturbation steers the two models toward different regions of the loss landscape: the stability branch is guided in directions that encourage alignment with previous knowledge, while the plasticity branch is updated solely based on the current task. As a result, each model captures distinct and complementary information, facilitating effective knowledge disentanglement for fusion.

The perturbed weights are defined as:

$$\widetilde{W}_{1:(t-1)} = \widehat{W}_{1:(t-1)} + \epsilon_{\text{KD}}, \tag{9}$$

$$\widetilde{W}_t = \widehat{W}_t + \epsilon_{\text{ReID}}. \tag{10}$$

These sharpness-aware approximations of the original weights allow gradient computation at adversarial points in the loss landscape, guiding the models toward flatter and more generalizable solutions. We then compute the gradients at the perturbed points and perform gradient descent:

$$\vec{g}_s = \nabla \left( \mathcal{L}_{\text{ReID}} + \mathcal{L}_{\text{KD}} \right) (\widetilde{W}_{1:(t-1)}), \tag{11}$$

$$\vec{g}_p = \nabla \mathcal{L}_{\text{ReID}}(\widetilde{W}_t). \tag{12}$$

Here, $\mathcal{L}_{\text{KD}}$ is a knowledge distillation loss that encourages consistency between the current model and the previous task model, while $\mathcal{L}_{\text{ReID}}$ denotes the ReID loss, typically composed of cross-entropy and triplet loss. To clarify the roles of each gradient, we denote $\vec{g}_s$ as the stability gradient, which ensures the preservation of knowledge from previous tasks, and $\vec{g}_p$ as the plasticity gradient, which facilitates learning of new task-specific representations. The updates are then applied as follows:

$$\widehat{W}_{1:(t-1)} \leftarrow \widehat{W}_{1:(t-1)} - \alpha \cdot \vec{g}_s, \tag{13}$$

$$\widehat{W}_t \leftarrow \widehat{W}_t - \alpha \cdot \vec{g}_p. \tag{14}$$

### 4.3 Integration of Stability and Plasticity Models

After training on each task, we update the model parameters using a fixed-weight linear combination of the stability model $\widehat{W}_{1:(t-1)}$ and the plasticity model $\widehat{W}_t$:

$$W_{1:t} \leftarrow (1 - \lambda) \cdot \widehat{W}_{1:(t-1)} + \lambda \cdot \widehat{W}_t,$$

where $\lambda$ is a predefined interpolation weight. As illustrated in Figure 4, this interpolation guides the fused model $W_{1:t}$ toward a low-loss region between the two solutions, balancing knowledge retention and adaptation. The fusion is applied once after completing training on the current task, thereby avoiding unnecessary optimization interference between the two branches. The resulting fused model $W_{1:t}$ is then used to initialize both branches for the next task, enabling smooth knowledge fusion and preserving prior knowledge more effectively.

## 5 Experiments

### 5.1 Datasets and Evaluation Metrics

As in previous studies (Pu et al., 2021; Ge et al., 2022; Sun & Mu, 2022), we conducted extensive experiments on four widely adopted LReID benchmark datasets: Market1501 (Zheng et al., 2015), MSMT17 (Wei et al., 2018), CUHK-SYSU (Xiao et al., 2017), and CUHK03 (Li et al., 2014). Since DukeMTMC-reID (Zheng et al., 2017) was retracted by its authors, we excluded it from both the training and evaluation phases. To evaluate the generalization capability of the model beyond the training domains, we additionally assessed its performance on six widely used unseen datasets: VIPeR Gray & Tao (2008), PRID Hirzer et al. (2011), GRID Loy et al. (2010), iLIDS Branch (2006), CUHK01 Li et al. (2012), and CUHK02 Li & Wang (2013). For quantitative evaluation, we followed standard LReID protocols and reported the Cumulative Matching Characteristic (CMC) at Rank-1 accuracy and the mean Average Precision (mAP).

### 5.2 Implementation Details

For the baseline, we use ResNet50 (He et al., 2016), pretrained on ImageNet (Deng et al., 2009), as the backbone network. Each dataset is trained for 50 epochs using the SGD optimizer with a weight decay of $5 \times 10^{-4}$ and an initial learning rate of 0.008, following a linear warm-up for the first 10 epochs. Each LReID method is trained with the optimizer, hyperparameters, and number of epochs specified in its original implementation, ensuring a fair and consistent comparison across methods. To improve generalization, we adopt ASAM (Kwon et al., 2021), with a perturbation radius of $\rho = 2.0$. The baseline, used for ablation comparisons, consists of an LReID framework trained with the standard ReID loss and a cosine similarity-based distillation loss. Our method is implemented as a plug-in module, easily integrable into any LReID framework without altering its core components. In all experiments, we fix the interpolation weight at $\lambda = 0.7$, determined empirically, with details in Appendix E.

### 5.3 Modular Integration with Existing Methods

We evaluate the proposed exemplar-free method by integrating it into six state-of-the-art LReID frameworks: AKA (Pu et al., 2021), PatchKD (Sun & Mu, 2022), LSTKC (Xu et al., 2024b), DKP (Xu et al., 2024a), DASK (Xu et al., 2025a), and LSTKC++ (Xu et al., 2025b). Performance is assessed under two training orders: Order 1 (Market1501 → CUHK-SYSU → MSMT17 → CUHK03) and Order 2 (MSMT17 → Market1501 → CUHK-SYSU → CUHK03), with results presented in Table 1 and Table 3. Across both orders, the proposed method consistently improves Seen-Avg and Unseen-Avg scores for all baselines. While performance on individual datasets may fluctuate, particularly in earlier sessions, the overall trend shows steady or improved accuracy as training progresses. In particular, our method enhances generalization to unseen domains. For example, when integrated with the baseline AKA, it improves the Unseen-Avg by 6.8 mAP and the Seen-Avg by 8.5 mAP in Order 1. Even when combined with the latest state-of-the-art method LSTKC++, it yields gains of 2.6 mAP in Unseen-Avg and 1.8 mAP in Seen-Avg. Notably, with LSTKC, the proposed method achieves consistent improvements across all individual datasets in both orders, indicating strong compatibility with competitive LReID baselines. A similar trend is

Table 1: Performance evaluation in the training Order: Market1501 → CUHK-SYSU → MSMT17 → CUHK03. The performance on each dataset is measured by using the model after the training with the last dataset is over. Bold denotes best performance.

| Methods | Market1501 | | CUHK-SYSU | | MSMT17 | | CUHK03 | | Seen-Avg | | Unseen-Avg | |
|---|---|---|---|---|---|---|---|---|---|---|---|---|
| | mAP | Rank-1 | mAP | Rank-1 | mAP | Rank-1 | mAP | Rank-1 | mAP | Rank-1 | mAP | Rank-1 |
| AKA | **53.6** | **74.0** | 74.3 | 77.6 | 4.7 | 13.0 | 33.7 | 34.4 | 41.6 | 49.8 | 47.6 | 42.6 |
| + Ours | 46.7 | 67.8 | **81.6** | **83.7** | **10.8** | **26.0** | **61.4** | **63.4** | **50.1** | **60.2** | **60.6** | **54.4** |
| PatchKD | 77.7 | **90.1** | **78.6** | **81.4** | 7.1 | 18.1 | 41.5 | 42.5 | 51.2 | 58.0 | 53.5 | 47.8 |
| + Ours | 65.0 | 83.6 | 78.1 | 79.9 | **11.1** | **27.0** | **56.8** | **59.2** | **52.7** | **62.4** | **58.6** | **50.0** |
| LSTKC | 54.8 | 75.6 | 80.5 | 82.2 | 18.6 | 41.5 | 43.3 | 45.3 | 49.3 | 61.1 | 55.5 | 48.7 |
| + Ours | **57.1** | **77.5** | **84.0** | **86.0** | **20.1** | **42.8** | **48.6** | **50.0** | **52.4** | **64.1** | **60.2** | **53.3** |
| DKP | **62.6** | **82.8** | 83.9 | 85.8 | 17.2 | 37.2 | 39.1 | 38.8 | 50.7 | 61.1 | 57.7 | 50.2 |
| + Ours | 57.7 | 78.8 | **84.5** | **86.3** | **18.9** | **39.9** | **47.5** | **48.3** | **52.2** | **63.3** | **60.0** | **52.5** |
| DASK | **63.5** | **82.7** | 81.9 | 83.5 | 26.5 | **53.4** | 46.8 | 48.6 | 54.7 | 67.1 | 63.0 | 55.8 |
| + Ours | 63.0 | 81.8 | **83.8** | **85.3** | **26.7** | 52.5 | **56.2** | **58.3** | **57.4** | **69.5** | **65.2** | **58.8** |
| LSTKC++ | **66.7** | **84.0** | 86.3 | 87.7 | **21.5** | **42.8** | 45.4 | 44.9 | 55.0 | 64.8 | 61.8 | 55.9 |
| + Ours | 65.2 | 82.4 | **86.4** | **88.2** | 20.0 | 41.4 | **55.6** | **57.4** | **56.8** | **67.3** | **64.4** | **57.7** |

Table 2: Performance comparison between the baseline and proposed modules after training on the final task.

| Methods | Market1501 | | CUHK-SYSU | | MSMT17 | | CUHK03 | | Seen-Avg | | Unseen-Avg | |
|---|---|---|---|---|---|---|---|---|---|---|---|---|
| | mAP | Rank-1 | mAP | Rank-1 | mAP | Rank-1 | mAP | Rank-1 | mAP | Rank-1 | mAP | Rank-1 |
| Baseline | 48.7 | 70.5 | 83.0 | 85.4 | 15.4 | 34.6 | 52.1 | 53.2 | 49.8 | 60.9 | 58.4 | 51.4 |
| Model1 | **56.0** | **76.0** | 81.2 | 83.4 | 14.7 | 33.6 | 48.9 | 50.6 | 50.2 | 60.9 | 56.0 | 48.9 |
| Model2 | 45.2 | 68.7 | 80.5 | 83.0 | 15.9 | 36.4 | **56.7** | **58.4** | 49.6 | 61.6 | 57.6 | 51.0 |
| Model1+2 (Ours) | 54.1 | 75.1 | **82.4** | **84.4** | **16.8** | **37.5** | 53.1 | 54.6 | **51.6** | **62.9** | **58.6** | **51.8** |

observed in Order 2, further supporting the robustness and general applicability of our method. These results demonstrate that the proposed approach serves as an effective plug-in module, improving both stability and generalization across LReID frameworks.

Note that the key challenge in LReID is not maximizing performance on the oldest domains, but reducing catastrophic forgetting while maintaining generalization across continually arriving domains. In Table 1 and Table 3, the fused model may exhibit slightly lower performance on older datasets, even when the Seen-Avg and Unseen-Avg scores improve. This behavior reflects the stability–plasticity trade-off: applying SAM on $L_{\mathrm{KD}}$ for the stability branch and on $L_{\mathrm{ReID}}$ for the plasticity branch allocates capacity to both retention and adaptation, yielding higher average scores and stronger generalization performance as training proceeds.

## 5.4  ABLATION STUDY

This ablation study is based on training Order 1. The Baseline refers to a standard LReID framework with ReID and distillation losses.

**Effectiveness of the Proposed Modules.** Table 2 presents the performance of different model variants. Model1 and Model2 each activate only one of the two proposed branches. Model1 uses the stability branch, and Model2 employs the plasticity branch. These single-branch configurations offer partial advantages. Model1 performs better on earlier tasks, while Model2 performs better on later ones, but their overall performance across seen and unseen domains remains suboptimal. The baseline model naively applies ASAM to the total training loss, which includes both the distillation and ReID losses, and achieves competitive generalization, particularly on unseen domains, due to its flatness-aware optimization. In contrast, our method, Model1+2 (Ours), selectively applies SAM to the distillation loss and interpolates the two branches. This dual-branch structure and flatness-aware

Table 3: Performance evaluation in the training Order: MSMT17 → Market1501 → CUHK-SYSU → CUHK03.

| Methods | MSMT17 | | Market1501 | | CUHK-SYSU | | CUHK03 | | Seen-Avg | | Unseen-Avg | |
|---|---|---|---|---|---|---|---|---|---|---|---|---|
| | mAP | Rank-1 | mAP | Rank-1 | mAP | Rank-1 | mAP | Rank-1 | mAP | Rank-1 | mAP | Rank-1 |
| AKA | **13.5** | **30.4** | 36.6 | 58.6 | 78.6 | 81.0 | 39.4 | 40.3 | 42.5 | 52.5 | 54.6 | 48.4 |
| + Ours | 11.6 | 27.6 | **37.6** | **59.7** | **79.4** | **82.0** | **45.1** | **44.9** | **43.4** | **53.5** | **58.5** | **52.5** |
| PatchKD | 31.7 | **58.8** | 48.0 | 70.8 | **80.6** | **82.7** | 43.7 | 43.9 | **51.5** | **64.0** | 58.3 | 51.2 |
| + Ours | 18.9 | 42.2 | **51.6** | **72.8** | 79.2 | 81.0 | **56.1** | **58.9** | 51.5 | 63.2 | **58.8** | **53.4** |
| LSTKC | 16.8 | 37.1 | 55.7 | 76.8 | 82.8 | 84.5 | 44.5 | 45.9 | 50.0 | 61.1 | 56.7 | 50.1 |
| + Ours | **17.0** | **37.6** | **60.0** | **79.6** | **85.2** | **86.9** | **50.4** | **51.8** | **53.1** | **64.0** | **61.1** | **54.0** |
| DKP | **19.9** | **42.0** | 57.8 | 78.3 | 83.7 | 85.7 | 42.7 | 42.7 | 51.1 | 62.2 | 58.9 | 52.3 |
| + Ours | 16.6 | 36.6 | **58.0** | **78.6** | **84.5** | **86.3** | **50.1** | **50.6** | **52.3** | **63.0** | **60.5** | **53.7** |
| DASK | **25.4** | **52.7** | **69.6** | **86.2** | 84.1 | 85.3 | 47.2 | 48.4 | 56.6 | 68.2 | 64.7 | 57.8 |
| + Ours | 20.5 | 44.8 | 69.4 | 85.1 | **86.0** | **87.5** | **58.5** | **61.0** | **58.6** | **69.6** | **66.9** | **60.3** |
| LSTKC++ | **21.7** | **44.2** | **65.0** | **82.3** | 85.3 | 86.6 | 48.4 | 49.1 | 55.1 | 65.5 | 63.6 | 57.2 |
| + Ours | 18.5 | 39.9 | 64.1 | 82.2 | **87.0** | **88.4** | **54.9** | **57.2** | **56.1** | **66.9** | **64.4** | **57.7** |

Table 4: Effect of applying SAM to the stability branch depending on diverse losses.

| Stability SAM | Seen-Avg | | Unseen-Avg | |
|---|---|---|---|---|
| | mAP | Rank-1 | mAP | Rank-1 |
| $L_{\text{ReID}}$ | 50.3 | 61.8 | 58.0 | 51.2 |
| $L_{\text{KD}}$ | **51.6** | **62.9** | **58.6** | **51.8** |
| $L_{\text{ReID}} + L_{\text{KD}}$ | 50.4 | 61.9 | 58.0 | 51.2 |

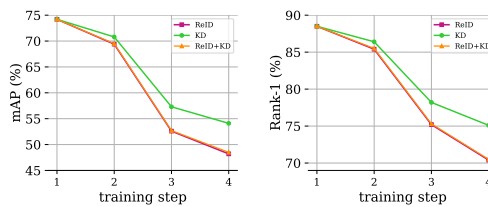

Figure 6: Effect of applying SAM to the stability branch on forgetting performance of the first dataset throughout training.

optimization enable more effective knowledge retention and adaptability, resulting in the highest average accuracy across all domains.

**Application of SAM to Various Losses.** Within our Dual-Model Selective SAM Training framework, we fix the plasticity branch applying SAM to $L_{\text{ReID}}$. To investigate the effect of applying SAM to different losses, we modify the stability branch by applying SAM to different losses: $L_{\text{ReID}}$, $L_{\text{KD}}$, and the total $L_{\text{ReID}}+L_{\text{KD}}$. Table 4 shows that applying SAM on $L_{\text{KD}}$ yields the best performance in terms of the Seen-Avg and Unseen-Avg scores, and Fig. 6 also shows lower forgetting performance across training. This outcome arises because the SAM perturbation for the total loss follows the gradient of $L_{\text{ReID}}+L_{\text{KD}}$, which is the sum of the two gradients. The two objectives have different roles, $L_{\text{ReID}}$ encourages current-task discrimination and $L_{\text{KD}}$ enforces alignment with the previous model, so their directions often conflict and the worst-case direction is dominated by the $L_{\text{ReID}}$ component, which leads to forgetting similar to when SAM is applied only to $L_{\text{ReID}}$. Applying SAM only to $L_{\text{KD}}$ is therefore most effective for building a stability branch that preserves prior knowledge.

## 6 CONCLUSION

In this paper, we proposed a novel continual learning framework for lifelong person re-identification (LReID) that integrates flat-minima optimization, knowledge distillation, and model interpolation. Inspired by recent advances in continual learning, our method guides optimization toward flat and robust regions of the loss landscape, improving generalization and reducing catastrophic forgetting. We selectively apply Sharpness-Aware Minimization (SAM) to the distillation loss of the stability model, promoting knowledge retention without hindering adaptability. After each task, we interpolate the weights of the stability and plasticity models to form a unified model, facilitating consistent knowledge accumulation. Extensive experiments show that our approach outperforms existing baselines on seen domains and significantly enhances generalization to unseen domains. Moreover, the proposed method is model-agnostic and easily applicable to existing LReID frameworks.

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

# APPENDIX

LLMs were used to aid in polishing the writing, improving clarity and readability.

## A FLAT MINIMA GUIDED WEIGHT INTERPOLATION ALGORITHM

The full training procedure of our framework is summarized in Algorithm 1. For completeness, we provide the explicit steps for selective SAM perturbation, dual-model optimization across stability and plasticity branches, and the flat-minima guided interpolation used to produce the fused model after each task. This algorithmic description complements the main text by outlining the exact update rules used in our implementation.

---

**Algorithm 1** Flat-Minima Guided Weight Interpolation

---

**Input**: Datasets $\{\mathcal{D}_1, \mathcal{D}_2, \ldots, \mathcal{D}_T\}$, model $W$, learning rate $\alpha$, radius $\rho$, total epochs $E$
**Output**: Trained model $W_{1:T}$

  Initialize $W_{1:1}$ randomly
  **for** epoch $e = 1$ to $E$ **do**
    Compute perturbation: $\boldsymbol{\epsilon} \leftarrow \rho \cdot \frac{\nabla \mathcal{L}_{\text{ReID}}(W_{1:1})}{\|\nabla \mathcal{L}_{\text{ReID}}(W_{1:1})\|_2}$
    Compute sharpness-aware gradient: $\vec{g} \leftarrow \nabla \mathcal{L}_{\text{ReID}}(W_{1:1} + \boldsymbol{\epsilon})$
    Update: $W_{1:1} \leftarrow W_{1:1} - \alpha \cdot \vec{g}$
  **end for**
  **for** $t = 2$ to $T$ **do**
    Initialize dual models: $\widehat{W}_{1:(t-1)} \leftarrow W_{1:(t-1)}, \widehat{W}_t \leftarrow W_{1:(t-1)}$
    **for** epoch $e = 1$ to $E$ **do**
      Sample mini-batch $(X, Y)$ from $\mathcal{D}_t$
      **SAM Perturbation**
      Compute $\vec{g}_{\text{KD}} \leftarrow \nabla \mathcal{L}_{\text{KD}}(\widehat{W}_{1:(t-1)}), \boldsymbol{\epsilon}_{1:(t-1)} \leftarrow \rho \cdot \frac{\vec{g}_{\text{KD}}}{\|\vec{g}_{\text{KD}}\|_2}$
      Compute $\vec{g}_{\text{ReID}} \leftarrow \nabla \mathcal{L}_{\text{ReID}}(\widehat{W}_t), \boldsymbol{\epsilon}_t \leftarrow \rho \cdot \frac{\vec{g}_{\text{ReID}}}{\|\vec{g}_{\text{ReID}}\|_2}$
      $\widetilde{W}_{1:(t-1)} \leftarrow \widehat{W}_{1:(t-1)} + \boldsymbol{\epsilon}_{1:(t-1)}, \widetilde{W}_t \leftarrow \widehat{W}_t + \boldsymbol{\epsilon}_t$
      **SAM Descent and Model Update**
      Compute stability gradient: $\vec{g}_s \leftarrow \nabla(\mathcal{L}_{\text{ReID}} + \mathcal{L}_{\text{KD}})(\widetilde{W}_{1:(t-1)})$
      Compute plasticity gradient: $\vec{g}_p \leftarrow \nabla \mathcal{L}_{\text{ReID}}(\widetilde{W}_t)$
      Update: $\widehat{W}_{1:(t-1)} \leftarrow \widehat{W}_{1:(t-1)} - \alpha \cdot \vec{g}_s, \widehat{W}_t \leftarrow \widehat{W}_t - \alpha \cdot \vec{g}_p$
    **end for**
    **Model fusion after training task** $t$: $W_{1:t} \leftarrow (1 - \lambda) \cdot \widehat{W}_{1:(t-1)} + \lambda \cdot \widehat{W}_t$
  **end for**
  **return** $W_{1:T}$

---

## B ADAPTIVE SHARPNESS-AWARE MINIMIZATION (ASAM)

Adaptive Sharpness-Aware Minimization (ASAM) (Kwon et al., 2021) extends SAM by making the perturbation scale-aware. Instead of treating all parameters uniformly, ASAM adjusts the perturbation direction based on the relative scale of each parameter. The perturbation is computed as:

$$\boldsymbol{\epsilon} = \rho \cdot \frac{\nabla_w \mathcal{L}(w)/(|w| + \eta)}{\|\nabla_w \mathcal{L}(w)/(|w| + \eta)\|_2}, \tag{15}$$

where $\eta$ is a small positive constant added for numerical stability. The term $(|w| + \eta)$ ensures that smaller-magnitude weights are not perturbed excessively, thereby enhancing scale-invariance and ensuring more stable optimization.

Once the perturbation is obtained, both SAM and ASAM update the weights using the gradient computed at the perturbed location:

$$w \leftarrow w - \alpha \cdot \nabla_w \mathcal{L}(w + \boldsymbol{\epsilon}), \tag{16}$$

where $\alpha$ denotes the learning rate. This two-step update encourages the model to converge to flatter and more generalizable minima.

## C Comparison between SAM and ASAM

Table 5: Average mAP and Rank-1 accuracy of the baseline LReID model trained with SAM and ASAM.

| Methods | Seen-Avg | | Unseen-Avg | |
|---|---|---|---|---|
| | mAP | Rank-1 | mAP | Rank-1 |
| SAM | 50.7 | 62.2 | 57.3 | 50.1 |
| ASAM | **51.6** | **62.9** | **58.6** | **51.8** |

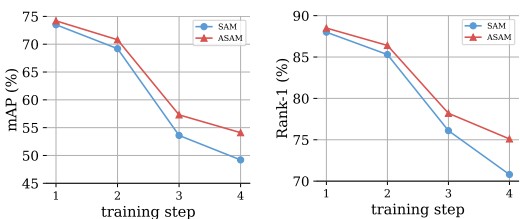

Figure 7: Evaluation of the non-forgetting performance on the first dataset at each training step under SAM and ASAM.

As described in the main paper, we adopt ASAM in place of standard SAM, as the normalization term in ASAM leads to more balanced gradient perturbations across layers with heterogeneous parameter scales. This property is particularly useful in our dual-model setup, where the stability and plasticity branches are guided by different loss functions and may exhibit different sensitivities to sharpness.

To validate the effectiveness of ASAM in this setting, we compare it to standard SAM under identical training conditions on the Order 1 training sequence. Figure 7 visualizes the non-forgetting performance on the first dataset at each training step, showing that ASAM better preserves earlier knowledge, especially in terms of non-forgetting. Table 5 summarizes the final mAP and Rank-1 accuracy across both seen and unseen benchmarks. ASAM consistently outperforms SAM in all metrics, demonstrating that input-aware perturbation is more effective for robust optimization in LReID. These results support our design choice of using ASAM throughout the main experiments.

Table 6: Performance comparison between the baseline and proposed modules after training on the final task.

| Methods | Market1501 | | CUHK-SYSU | | MSMT17 | | CUHK03 | | Seen-Avg | | Unseen-Avg | |
|---|---|---|---|---|---|---|---|---|---|---|---|---|
| | mAP | Rank-1 | mAP | Rank-1 | mAP | Rank-1 | mAP | Rank-1 | mAP | Rank-1 | mAP | Rank-1 |
| Baseline | 46.2 | 67.9 | 82.2 | 84.3 | 14.9 | 33.3 | 51.2 | 53.2 | 48.6 | 59.7 | 56.8 | 49.9 |
| Model1 | **54.4** | **75.0** | 81.5 | 83.5 | 15.4 | 34.2 | 50.2 | 52.1 | 50.4 | 61.2 | 55.6 | 48.6 |
| Model2 | 43.5 | 66.2 | 80.4 | 82.6 | 15.9 | 35.8 | **55.8** | **57.6** | 48.9 | 60.5 | 56.8 | 49.7 |
| Model1+2 (Ours) | 49.2 | 70.8 | **82.1** | **84.1** | **16.9** | **37.2** | 54.6 | 56.6 | **50.7** | **62.2** | **57.3** | **50.1** |

As shown in Table 6, comparable improvements were also observed when using standard SAM instead of ASAM, indicating that our proposed modules are consistently effective across different sharpness-aware optimization strategies.

## D Effect of ASAM on Existing LReID Baseliness

To clarify the effect of the optimizer, we additionally provide results under the ASAM setting in Tables 7 and 8. Applying ASAM alone often improves the performance of existing LReID methods, yet this trend is not universal. For example, DKP exhibits a clear performance drop when trained with ASAM, indicating that ASAM does not consistently benefit all baselines. For this reason, and to ensure a fair comparison, the main paper reports the baseline results trained with their original optimizer (e.g, SGD or Adam), while the proposed method is evaluated with ASAM as described in our approach.

When comparing ASAM with ASAM + our method, our framework still provides consistent performance gains across baselines. Notably, DKP, which suffers performance degradation under ASAM alone, recovers and improves once our framework is applied. Although the improvements do not always increase both seen and unseen averages simultaneously, the proposed method achieves a more balanced trade-off and reduces the bias that ASAM alone may introduce.

Table 7: Performance evaluation in the training order: Market1501 → CUHK-SYSU → MSMT17 → CUHK03, under the ASAM setting. "+ Ours" denotes ASAM + our framework.

| Methods | Market1501 | | CUHK-SYSU | | MSMT17 | | CUHK03 | | Seen-Avg | | Unseen-Avg | |
|---|---|---|---|---|---|---|---|---|---|---|---|---|
| | mAP | Rank-1 | mAP | Rank-1 | mAP | Rank-1 | mAP | Rank-1 | mAP | Rank-1 | mAP | Rank-1 |
| AKA (ASAM) | **54.3** | **73.7** | 77.6 | 80.4 | 5.7 | 15.5 | 38.4 | 37.7 | 44.0 | 51.8 | 54.9 | 48.5 |
| + Ours | 46.7 | 67.8 | **81.6** | **83.7** | **10.8** | **26.0** | **61.4** | **63.4** | **50.1** | **60.2** | **60.6** | **54.4** |
| PatchKD (ASAM) | **77.7** | **90.4** | **81.0** | **83.1** | 7.6 | 19.1 | 45.8 | 46.4 | **53.0** | **59.7** | **56.7** | 49.8 |
| + Ours | 65.0 | 83.6 | 78.1 | 79.9 | **11.1** | **27.0** | **56.8** | **59.2** | 52.7 | 62.4 | 58.6 | **50.0** |
| LSTKC (ASAM) | 55.5 | 76.6 | **84.6** | **86.8** | 19.3 | 41.6 | 46.5 | 47.9 | 51.5 | 63.2 | 60.0 | **53.1** |
| + Ours | **57.1** | **77.5** | 84.0 | 86.0 | **20.1** | **42.8** | **48.6** | **50.0** | **52.4** | **64.1** | **60.2** | 53.3 |
| DKP (ASAM) | **61.1** | **80.0** | **84.8** | **86.3** | 16.0 | 35.6 | 36.1 | 34.9 | 49.5 | 59.2 | 59.4 | 52.1 |
| + Ours | 57.7 | 78.8 | 84.5 | 86.3 | **18.9** | **39.9** | **47.5** | **48.3** | **52.2** | **63.3** | **60.0** | **52.5** |
| DASK (ASAM) | **63.0** | **82.1** | **84.2** | **86.1** | 26.2 | **53.4** | 51.0 | 51.9 | 56.1 | 68.4 | **65.8** | **59.1** |
| + Ours | 63.0 | 81.8 | 83.8 | 85.3 | **26.7** | 52.5 | **56.2** | **58.3** | **57.4** | **69.5** | 65.2 | 58.8 |
| LSTKC++ (ASAM) | **66.5** | **83.7** | **87.5** | **89.0** | **20.2** | 41.2 | 47.4 | 47.6 | 55.4 | 65.4 | **65.7** | **59.5** |
| + Ours | 65.2 | 82.4 | 86.4 | 88.2 | 20.0 | **41.4** | **55.6** | **57.4** | **56.8** | **67.3** | 64.4 | 57.7 |

Table 8: Performance evaluation in the training Order: MSMT17 → Market1501 → CUHK-SYSU → CUHK03.

| Methods | MSMT17 | | Market1501 | | CUHK-SYSU | | CUHK03 | | Seen-Avg | | Unseen-Avg | |
|---|---|---|---|---|---|---|---|---|---|---|---|---|
| | mAP | Rank-1 | mAP | Rank-1 | mAP | Rank-1 | mAP | Rank-1 | mAP | Rank-1 | mAP | Rank-1 |
| AKA (ASAM) | **13.1** | **30.3** | **42.9** | **64.6** | **81.2** | **83.9** | 39.4 | 39.7 | **44.1** | **54.6** | 57.1 | 50.8 |
| + Ours | 11.6 | 27.6 | 37.6 | 59.7 | 79.4 | 82.0 | **45.1** | **44.9** | 43.4 | 53.5 | **58.5** | **52.5** |
| PatchKD (ASAM) | **27.3** | **52.6** | 49.3 | 71.1 | **82.7** | **84.6** | 50.2 | 53.0 | **52.9** | **65.3** | **61.2** | 55.1 |
| + Ours | 18.9 | 42.2 | **51.6** | **72.8** | 79.2 | 81.0 | **56.1** | **58.9** | 51.5 | 63.2 | 58.8 | **53.4** |
| LSTKC (ASAM) | 16.6 | 36.9 | 57.5 | 78.0 | **85.6** | **87.3** | 47.0 | 47.9 | 51.7 | 62.5 | 61.0 | **54.3** |
| + Ours | **17.0** | **37.6** | **60.0** | **79.6** | 85.2 | 86.9 | **50.4** | **51.8** | **53.1** | **64.0** | **61.1** | 54.0 |
| DKP (ASAM) | **18.0** | **38.9** | 54.4 | 75.0 | **84.9** | **86.7** | 36.2 | 35.9 | 48.4 | 59.1 | 59.3 | 52.5 |
| + Ours | 16.6 | 36.6 | **58.0** | **78.6** | 84.5 | 86.3 | **50.1** | **50.6** | **52.3** | **63.0** | **60.5** | **53.7** |
| DASK (ASAM) | **22.6** | **48.9** | **69.8** | **86.1** | **86.4** | **88.1** | 54.8 | 56.9 | 58.4 | **70.0** | **67.5** | **61.4** |
| + Ours | 20.5 | 44.8 | 69.4 | 85.1 | 86.0 | 87.5 | **58.5** | **61.0** | **58.6** | 69.6 | 66.9 | 60.3 |
| LSTKC++ (ASAM) | **20.3** | **42.4** | 61.8 | 79.5 | **87.3** | **88.7** | 46.5 | 46.9 | 54.0 | 64.3 | **65.6** | **59.6** |
| + Ours | 18.5 | 39.9 | **64.1** | **82.2** | 87.0 | 88.4 | **54.9** | **57.2** | **56.1** | **66.9** | 64.4 | 57.7 |

# E  EFFECT OF PARAMETER $\lambda$

Table 9: Effect of the interpolation weight $\lambda$ in the linear model fusion on average mAP and Rank-1 accuracy.

| $\lambda$ | Seen-Avg | | Unseen-Avg | |
|---|---|---|---|---|
| | mAP | Rank-1 | mAP | Rank-1 |
| 0.3 | 50.8 | 62.3 | 55.0 | 51.4 |
| 0.5 | 51.4 | 62.7 | 58.3 | 51.6 |
| 0.7 | **51.6** | **62.9** | **58.6** | **51.8** |
| 0.9 | 50.8 | 61.8 | 57.0 | 49.8 |

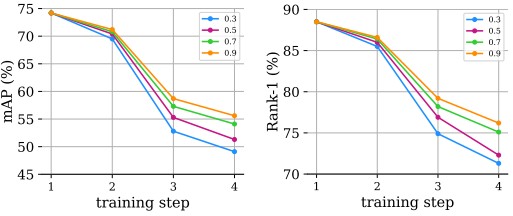

Figure 8: Effect of the interpolation weight $\lambda$ on the non-forgetting performance of the first dataset throughout training.

We investigate the impact of the interpolation weight $\lambda$ used for linear fusion of the dual branches. As shown in Figure 8 and Table 9, varying $\lambda$ significantly affects the trade-off between preserving performance on the first dataset and achieving high accuracy across seen and unseen domains. Since the first dataset is typically the most affected by forgetting, its accuracy serves as a measure of knowledge preservation and provides insights into the non-forgetting. As $\lambda$ decreases, forgetting becomes more pronounced, leading to a clear performance drop on the first dataset. In contrast, $\lambda = 0.9$ best preserves earlier knowledge but leads to less adaptability, resulting in lower average accuracy. The setting $\lambda = 0.7$ offers the most balanced outcome, yielding the highest average performance across all domains. Based on these observations, we adopt $\lambda = 0.7$ for all subsequent experiments.

### E.1 ADDITIONAL VALIDATION OF THE INTERPOLATION WEIGHT $\lambda$

Table 10: Effect of $\lambda$ under the DASK.

| $\lambda$ | Seen-Avg | | Unseen-Avg | |
|---|---|---|---|---|
| | mAP | Rank-1 | mAP | Rank-1 |
| 0.5 | 54.6 | 66.8 | 64.0 | 57.7 |
| 0.7 | **57.4** | **69.5** | 65.2 | 58.8 |
| 0.9 | 56.7 | 68.7 | **66.3** | **59.7** |

Table 11: Effect of $\lambda$ under the LSTKC++.

| $\lambda$ | Seen-Avg | | Unseen-Avg | |
|---|---|---|---|---|
| | mAP | Rank-1 | mAP | Rank-1 |
| 0.5 | 55.1 | 65.8 | 61.3 | 54.6 |
| 0.7 | **56.8** | **67.3** | **64.4** | **57.7** |
| 0.9 | 54.6 | 65.2 | 60.0 | 52.7 |

We additionally evaluate the interpolation weight $\lambda$ under two alternative LReID settings, DASK and LSTKC++, to verify whether a universal choice is appropriate. As shown in Tables 10 and 11, $\lambda = 0.7$ yields the highest overall performance in both settings. Although DASK shows slightly higher Unseen-Avg performance at $\lambda = 0.9$, this comes at the cost of a noticeable drop on seen domains. In contrast, $\lambda = 0.7$ provides the most effective consolidation of the stability and plasticity branches, achieving the best balance between retaining earlier knowledge and adapting to new domains. These results support the use of a single interpolation weight across different LReID methods.

## F RESULTS WITH DIFFERENT TRAINING ORDERS

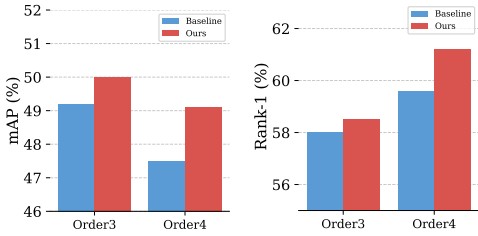

Figure 9: Comparison of mAP and Rank-1 performance under two different training orders.

To assess the robustness of our method under varying domain sequences, we additionally experimented with two training orders: Order 3 (CUHK03 $\rightarrow$ MSMT17 $\rightarrow$ CUHK-SYSU $\rightarrow$ Market1501) and Order 4 (CUHK03 $\rightarrow$ CUHK-SYSU $\rightarrow$ Market1501 $\rightarrow$ MSMT17). As shown in Figure 9, the proposed method consistently demonstrated performance improvements in both mAP and Rank-1, confirming its effectiveness across different task permutations.

## G AVERAGE PERFORMANCE ACROSS TRAINING STEPS

Figure 10 presents the average mAP and Rank-1 accuracy at each training step, evaluated on the dataset learned at that step under training Order 1. For all LReID baselines, incorporating our framework consistently improves step-wise performance, indicating that the proposed method enhances both stability on earlier tasks and adaptability to newly introduced data throughout the training sequence.

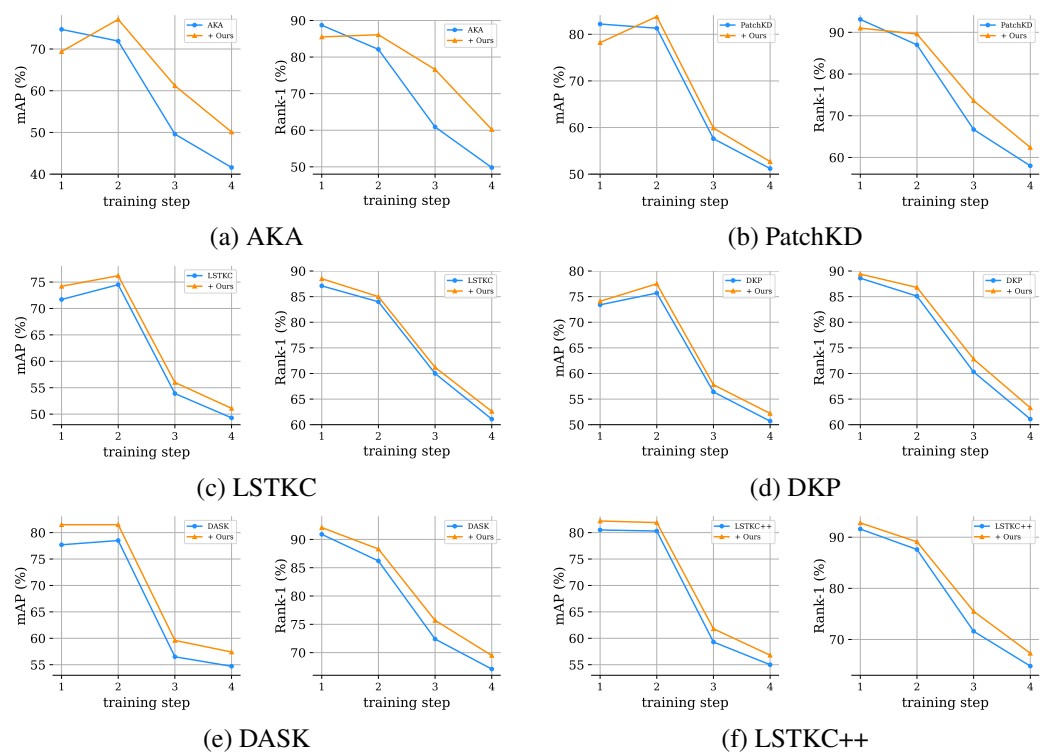

Figure 10: Average mAP and Rank-1 across training steps under Training Order 1, evaluated as the average performance on the dataset learned at each step, comparing each LReID method with and without the proposed framework.

## H DATASETS ANALYSIS

Table 12: Statistics of seen and unseen LReID datasets used in our experiments.

| Type | Dataset | Scale | #train IDs | #test IDs |
|---|---|---|---|---|
| Seen | Market1501 (Zheng et al., 2015) | large | 500 (751) | 750 |
| | MSMT17 (Wei et al., 2018) | large | 500 (1041) | 3060 |
| | CUHK-SYSU (Xiao et al., 2017) | mid | 500 (942) | 2900 |
| | CUHK03 (Li et al., 2014) | mid | 500 (767) | 700 |
| Unseen | VIPeR (Gray & Tao, 2008) | small | – | 316 |
| | PRID (Hirzer et al., 2011) | small | – | 649 |
| | GRID (Loy et al., 2010) | small | – | 126 |
| | iLIDS (Branch, 2006) | small | – | 60 |
| | CUHK01 (Li et al., 2012) | small | – | 486 |
| | CUHK02 (Li & Wang, 2013) | mid | – | 239 |

Table 12 reports the detailed statistics and properties of the benchmark datasets used in our experiments. For training, we uniformly sample 500 identities from each seen benchmark dataset, following the standard protocol adopted in prior LReID works Pu et al. (2021); Sun & Mu (2022). Unseen datasets are used solely for evaluation to assess the generalization performance of the model across diverse domains. The seen datasets include Market1501 Zheng et al. (2015), MSMT17 Wei et al. (2018), CUHK-SYSU Xiao et al. (2017), and CUHK03 Li et al. (2014), while the unseen datasets consist of VIPeR Gray & Tao (2008), PRID Hirzer et al. (2011), GRID Loy et al. (2010), iLIDS Branch (2006), CUHK01 Li et al. (2012), and CUHK02 Li & Wang (2013).

# I  DETAILED ANALYSIS OF FIGURES 2 AND 3

Figures 2 and 3 in the main paper provide a detailed comparison between SAM Foret et al. (2020) and standard SGD in the context of LReID training. Both figures are based on a baseline ReID framework trained solely with the standard ReID loss, which includes cross-entropy and triplet loss, without any additional lifelong learning components. For SAM, we use the ASAM Kwon et al. (2021) variant with the perturbation radius set to $\rho = 2.0$, following the same configuration as the main paper.

Figure 2 illustrates the loss landscape after continual finetuning, where the model is first trained on the Market1501 dataset and then adapted to CUHK-SYSU. The visualization demonstrates that SAM yields flatter minima than SGD even after the first task, and this difference becomes more pronounced after the second task. These flatter solutions correlate strongly with enhanced generalization and reduced forgetting. Figure 3 presents the non-forgetting and generalization performance across tasks under the Order 1 training sequence. The SAM-based model consistently outperforms the SGD-based model on both seen and unseen domains. These results underscore the importance of flat-minima optimization in enhancing generalization and mitigating forgetting in LReID.

