# OpenReview forum: "Towards Better Generalization in Lifelong Person Re-Identification with Flatness-Aware Learning"
_ICLR.cc/2026/Conference — Submitted to ICLR 2026_

### Official Review · Reviewer_3Ukf · 2025-10-25

**Soundness:** 2
**Presentation:** 2
**Contribution:** 2
**Rating:** 4
**Confidence:** 5

**Summary:**

This paper addresses the challenge of lifelong person re-identification (LReID), where models sequentially adapt to new domains while avoiding catastrophic forgetting. The proposed approach unifies three main components: (1) selective Sharpness-Aware Minimization (SAM) applied to only the knowledge distillation loss, (2) dual-model training where stability and plasticity branches are optimized independently, and (3) interpolation of these two models’ weights for a fused model that balances retention and adaptability. Experiments and ablation studies on LReID benchmarks demonstrate consistent improvements in generalization and knowledge retention over a range of state-of-the-art baselines.

**Strengths:**

1. The application of SAM to the distillation branch provides a fresh angle within LReID. The connection to generalization and robustness is clearly supported by the visualizations in Figure 2 and Figure 3.
2. The proposed method is easily embedded into various existing LReID architectures, as shown by empirical integration with six state-of-the-art baselines.

**Weaknesses:**

1. While selective SAM to the distillation loss is novel in this context, the core ideas, dual-branch training and linear weight interpolation, are not entirely new and can be viewed as straightforward extensions of existing regularization and model fusion paradigms (Exponential Moving Average in DKP, DASK).
2. The ablation in Table 2 and Table explores various losses for SAM, but does not consider alternative fusion approaches, such as non-linear, confidence-weighted, or meta-learned combinations of the two branches. Given that linear weight interpolation is a central design choice (Section 4.3 and Figure 4), the lack of comparison with more advanced model merging strategies leaves a gap in validating the optimality of their method.
3. The method’s reliance on dual-model maintenance and per-branch optimization likely increases both memory and computation costs compared to standard single-model baselines, but no discussion or empirical measurement of these costs is provided.
4. The fixed hyperparameter $\lambda$ for model fusion is justified with an ablation in the appendix, but the rationale for choosing a universal value across all settings is limited.

**Questions:**

The method requires maintaining two full network. Can you provide measurements or analysis of computational and memory overhead (training time, inference speedup/slowdown, GPU memory footprint) relative to single-model baselines?

---

> ### Author Response · Authors · 2025-11-20
>
> **1. Difference from existing regularization and model fusion paradigms**
>
> The dual branch structures and parameter interpolation in the proposed method are fundamentally different from existing regularization or fusion techniques in terms of purpose and mechanism. Although prior LReID methods such as DKP and DASK employ EMA style updates, these approaches simply maintain a smoothed trajectory of a single model. In contrast, our dual model design assigns explicitly differentiated roles to the two branches which operate in a complementary manner with selective SAM and the interpolation step. One branch preserves stable knowledge from previous tasks, whereas the other deliberately explores flatter and more adaptable regions under the guidance of the distillation gradient. The linear interpolation is not used as a generic smoothing operation but as a flatness aware consolidation step that prevents incompatibility between the two optimization trajectories. This coordinated interaction among selective SAM, dual trajectory training, and interpolation is unique to our framework and is essential for achieving stability without replay buffers or architectural modification. We will discuss and clarify this in the revised manuscript. Please see the $2^{\text{nd}}$ paragraph in Section 4.
>
> **2. Comparison of alternative model fusion methods**
>
> Thanks for this insightful comment. Our aim is to design a lightweight, parameter free, and architecture independent consolidation step that can be seamlessly integrated into existing LReID pipelines without adding extra modules or training objectives. Therefore, we did not consider sophisticated mechanisms such as nonlinear, confidence weighted, or meta learned combinations.
>
> **3. Discussion on memory and computation costs**
>
> As the reviewer suggested, we reported the computational and memory overhead of our dual model training. In LReID setting, the inference cost remains identical to the baseline because only the interpolated model is used at test time. Both methods therefore share the same inference latency and GPU memory footprint. During the training on an RTX 3090 GPU, the baseline requires 3.98 hours, whereas our method requires 6.28 hours, reflecting 57.7 percent increase due to the dual model optimization and the selective SAM perturbation. The peak GPU memory usage increases moderately from 9062–9858 MB (baseline) to 9062–10518 MB (ours), corresponding to an overhead of approximately 6 percent to 7 percent. Note that these costs arise during training only, while inference efficiency remains unchanged. We consider this overhead acceptable given the improved stability and performance in the lifelong learning setting, particularly because our approach is replay free and can be seamlessly plugged into existing frameworks without requiring any architectural modification.
>
> **4. Rationale of hyperparameter setting**
>
> We clarified the hyperparameter setting. We tuned λ on the main backbone and observed that the performance curve remains stable around λ=0.7 for both seen and unseen datasets. To maintain the fairness and avoid unintentionally favoring specific plug-in baselines through per method hyperparameter tuning, we intentionally kept λ identical for all plug-in experiments. In the revised version, we additionally evaluate λ under two other LReID methods to verify whether the same conclusion holds beyond the main backbone. The results, reported in Appendix E.1, show that the performance trend remains consistent across these methods, which supports our decision to adopt a single universal value for λ across all settings. This will be clarified in the revised manuscript.

---

### Official Review · Reviewer_i8z3 · 2025-10-26

**Soundness:** 2
**Presentation:** 2
**Contribution:** 2
**Rating:** 4
**Confidence:** 5

**Summary:**

This paper addresses the lifelong person reidentification (LReID) task, which has been extensively investigated recently. This paper aims to unify selective flatness-aware optimization, dual-model training, and model interpolation. Promising performance is achieved compared to existing works.

**Strengths:**

1. This paper is well-structured and smoothly written.

2. Promising performance is achieved compared to existing works, verifying the effectiveness of the proposed framework.

**Weaknesses:**

1. Unclear motivation. This paper does not explain the necessity of unifying selective flatness-aware optimization,  dual-model training, and model interpolation.

2. Limited analysis of the relation with the LReID task. This paper introduces an LReID method, while the ReID-relevant loss is not introduced.

3. Limited illustration. This paper does not contain the framework figure containing the main data stream, making it hard for readers to understand some key designs, such as dual-model training and model interpolation.

4. Unfair comparison. The training setting of this paper is different from the previous papers, where an unusual optimizer, ASDM, is adopted. Therefore, it is unclear whether the improvement in this paper is achieved via training setting bias compared to the existing works.

**Questions:**

Please refer to the weakness.

---

> ### Author Response · Authors · 2025-11-20
>
> **1. Clarification of motivation**
>
> Thanks for this constructive comment. When SAM is applied only to the KD loss of the stability branch, the resulting stability model retains knowledge but becomes less adaptable. The mAP performance on CUHK03 decreases from 52.1 of the baseline to 48.9, and its Unseen-Avg mAP decreases from 58.4 to 56.0, showing that applying SAM only to the distillation objective cannot prevent reduced generalization to new domains.  The plasticity model, trained with only the ReID loss and SAM applied to ReID, adapts more strongly to the new domain but loses stability. The mAP performance on Market1501 drops from 48.7 of the baseline to 45.2, and the Seen-Avg mAP decreases from 49.8 to 49.6, indicating that the plasticity branch alone cannot preserve previously learned knowledge. In contrast, fusing the two branches yields clear improvements, raising the Seen-Avg mAP from 49.8 to 51.6 and the Unseen-Avg mAP from 58.4 to 58.6. These results demonstrate that selective SAM alone cannot prevent the drift in the plasticity direction, the plasticity branch alone cannot retain prior knowledge, and interpolation alone cannot correct sharp or unstable regions. Their unified use is essential for achieving a stable balance between adaptation and retention in LReID. We clarified the motivation in the revised manuscript. Please see the $2^{\text{nd}}$ paragraph in Section 4 and Table 2.
>
> **2. Limited analysis of LReID task**
>
> We clarify that the ReID loss is essential in LReID because it provides direct supervision for learning identity-discriminative features on the new domain, whereas the distillation loss preserves knowledge from previous domains. The two losses serve complementary roles: ReID loss enables adaptation to new identities, and distillation prevents forgetting. In the revised manuscript, we explicitly describe this relationship and add citations to prior LReID works (Sun & Mu, 2022; Xu et al., 2024b) to make this connection clear. Please refer to Lines 199~201 in Section 3.1 of the revised manuscript.
>
>
> **3. Illustration of whole framework**
>
> As the reviewer suggested, we have added a figure of the unified framework that illustrates the full data flow, including the dual model training process, in the revised manuscript. We also clarified that selective SAM, dual branch optimization, and model interpolation operate as a single integrated pipeline rather than three independent modules. Please see the Figure 5.
>
> **4. Unfair comparison**
>
> We emphasize that unifying all methods under ASAM does not provide a fair comparison. Some LReID methods, such as DKP, suffer clear performance degradation when trained with ASAM, so forcing all baselines to use ASAM would unfairly disadvantage certain methods. Therefore, we follow each baseline’s original optimizer (e.g., SGD or Adam) to ensure a faithful comparison. For the ablation results in Table 2, we additionally train all variants with ASAM to evaluate each proposed component under a shared optimization setting. In this experiment, our method outperforms the ASAM-only baseline, showing that the improvements do not come from ASAM itself. To further address the reviewer’s concern, we retrain all baselines with ASAM and report the results in Appendix D. Across all methods, ASAM combined with our framework outperforms ASAM alone and produces a more balanced trade-off between seen and unseen domains. Notably, methods that degrade under ASAM recover and improve once our framework is applied. These results confirm that the performance gains are due to our method rather than the optimizer.

---

### Official Review · Reviewer_czw5 · 2025-10-31

**Soundness:** 2
**Presentation:** 2
**Contribution:** 2
**Rating:** 2
**Confidence:** 5

**Summary:**

This paper proposes an LReID method that unifies selective flatness-aware optimization, where a stability model trained with distillation loss retains prior knowledge, and a plasticity model optimized solely for the current domain. It further selectively applies Sharpness-Aware Minimization (SAM) only to the distillation loss, guiding the stability model toward flat and robust solutions.

**Strengths:**

1. The structure is complete.
2. The experimental results verify the effectiveness of the proposed method to some extent.

**Weaknesses:**

1. The motivation is unclear. I don't understand what the authors mean by "well-behaved regions of the loss landscape." Also, the definition of "sharp or incompatible solutions" is confusing. The authors should use the simplest language possible to explain their ideas.
2. The proposed method is not clear enough. Due to the lack of a diagram to illustrate the method, I am unclear about what the authors did and how the method works.
3. The lack of visualization experiments makes it difficult to intuitively understand why the proposed method leads to the final performance improvement.

**Questions:**

See the comments below.

---

> ### Author Response · Authors · 2025-11-20
>
> **1. Clarification of motivation**
>
> We appreciate the reviewer’s comments and sorry for unclear descriptions. Specifically, “well-behaved regions” refer to the areas where small changes in the parameters do not cause large performance drops, and “sharp or incompatible solutions” mean the solutions that fit the current task but significantly degrade the performance on previous tasks. We clarified the motivation in the revised manuscript. Please see the Lines 199~201 in Section 1.
>
> **2. Clarification of the proposed method**
>
> Following the reviewer’s comment, we have added a figure of the unified framework that illustrates the full data flow, including the dual model training process, in the revised manuscript. We also clarified that selective SAM, dual branch optimization, and model interpolation operate as a single integrated pipeline rather than three independent modules. Please see the $2^{\text{nd}}$ paragraph in Section 4 and Figure 5.
>
> **3. Lack of visualization experiments**
>
> As the reviewer suggested, we included additional visualization experiments in Appendix G, where we compared stepwise performance curves and illustrated the effect of the proposed framework across LReID methods. These visualizations clearly show how our method maintains higher accuracy on earlier tasks, while achieving competitive adaptation on newly introduced domains, offering direct evidence of its effectiveness.

---

### Official Review · Reviewer_AiGX · 2025-11-01

**Soundness:** 2
**Presentation:** 2
**Contribution:** 2
**Rating:** 4
**Confidence:** 4

**Summary:**

This paper addresses a novel framework for lifelong person re-identification
(LReID) task. The framework, which unifies selective flatness-aware optimization,
dual-model training, and model interpolation, achieves a promising performance and reduces catastrophic forgetting rates.

**Strengths:**

This paper originally generates a framework which can be easily interpolated and used.

This paper is written smoothly and does not have any long sentences which may lead to understanding difficulty.

**Weaknesses:**

Limited illustration. This paper does not have any figures that illustrate the whole framework, which includes the dual-model training. And this may lead to a misunderstanding about how the framework actually works.

Unmatched result. The results in Table 4 do not align with Table 2, as the experiment for both has the same configuration, while the results are very different. Also, there is a typo in Table 4, Line 3.

Limited formula. This paper provides limited formula derivations. Some formulas, such as the derivation of the selective SAM gradient, have not been given, which may lead to difficulties for readers to replicate.

Lack of consistent integration. This paper does not provide enough support for the integration of the main modules, which may make it seem like three modules instead of one framework.

**Questions:**

Please refer to the weaknesses.

---

> ### Author Response · Authors · 2025-11-20
>
> **1. Illustration of whole framework**
>
> We appreciate the reviewer’s feedback regarding the clarity of the overall framework. In the revised manuscript, we have added a figure of the unified framework that illustrates the full data flow, including the dual model training process. We also clarified that selective SAM, dual branch optimization, and model interpolation operate as a single integrated pipeline rather than three independent modules. Please see the $2^{\text{nd}}$ paragraph in Section 4 and Figure 5.
>
>
> **2. Unmatched result**
>
> Thanks for your careful reading. We apologize for the typo in Table 4 Line 3, which has been corrected as $L_{\text{ReID}}$ + $L_{\text{KD}}$ in the revised manuscript. However, we politely note that there is no misalignment between Table 2 and Table 4. Specifically, the results in the 2nd line of $L_{\text{KD}}$ in Table 4 exactly match the average performance of Model 1+2 (Ours) shown in the last line in Table 2.
>
>
> **3. Limited formula**
>
> Selective SAM does not alter the underlying optimization procedure of standard SAM; instead, it modifies which loss contributes to the perturbation. As clarified in Section 4.1, each branch computes its SAM perturbation using only the loss assigned to that branch, and the corresponding parameters are updated along that direction. The stability branch therefore follows the ReID loss, whereas the adaptive branch uses the distillation loss to explore flatter regions. Since the mathematical form of the SAM update remains unchanged from the original formulation, we refer the standard SAM paper for the full derivation. We will clarify this in the revised manuscript. Please see the $4^{\text{th}}$ paragraph in Section 3.2.

---

### Meta-Review · Area_Chair_Hgsc · 2025-12-26

**Summary:**

This paper explores the problem of person re-identification under continuous learning conditions. It investigates the relationship between Sharpness-Aware Minimization (SAM) and distillation learning from the perspectives of generalization learning and flatness learning. Experiments validate the effectiveness of the proposed method to some extent, demonstrating its stability and resistance to forgetting on lifelong learning ReID tasks.

**Reviewer Concerns:**

The initial draft of this paper did not include method diagrams; the revised draft added simple illustrations and training flowcharts. Reviewers AiGX pointed out issues such as spelling errors in the experimental results and limited derivations.

Furthermore, reviewer czw5 acknowledged the method to some extent but felt the visualization experiments were insufficient and the paper's motivation was unclear. The authors added a simple unified framework description and modified some keywords, but further clarification is needed regarding the theoretical analysis, the motivation for performance improvements, and the essential relationship with incremental learning and lifelong learning mechanisms. Because flatness-aware optimization has been widely applied to incremental learning and even incremental few-shot learning, the authors must delve deeper into its essential relationship with the subtasks currently being studied.

Furthermore, reviewers 3Ukf and i8z3 both expressed concerns regarding the core motivation, experimental comparisons, and innovativeness. Although the authors offered some responses, AC believes these responses lacked detailed analysis and theoretical or experimental support. Moreover, these weaknesses have been agreed upon by multiple reviewers.

**Reviewer Scores:**

In summary, this paper does indeed have numerous issues regarding methodological motivation, methodological process description, visualization experiments, theoretical analysis, and proof. The reviewers unanimously gave it a negative evaluation, and there is no strong evidence to support a change of opinion from the reviewers.

---

### Decision · Program_Chairs · 2026-01-26

Reject